# Delivery of Adeno-Associated Virus Vectors to the Central Nervous System for Correction of Single Gene Disorders

**DOI:** 10.3390/ijms25021050

**Published:** 2024-01-15

**Authors:** Rrita Daci, Terence R. Flotte

**Affiliations:** 1Department of Neurosurgery, University of Massachusetts Chan Medical School, 55 N Lake Ave, Worcester, MA 01655, USA; rrita.daci@umassmemorial.org; 2Horae Gene Therapy Center, University of Massachusetts Chan Medical School, 368 Plantation Street, Worcester, MA 01605, USA; 3Department of Pediatrics, University of Massachusetts Chan Medical School, 55 N Lake Ave, Worcester, MA 01655, USA

**Keywords:** gene therapy, adeno-associated virus vector, AAV, gene delivery, inherited disorders

## Abstract

Genetic disorders of the central nervous system (CNS) comprise a significant portion of disability in both children and adults. Several preclinical animal models have shown effective adeno-associated virus (AAV) mediated gene transfer for either treatment or prevention of autosomal recessive genetic disorders. Owing to the intricacy of the human CNS and the blood–brain barrier, it is difficult to deliver genes, particularly since the expression of any given gene may be required in a particular CNS structure or cell type at a specific time during development. In this review, we analyzed delivery methods for AAV-mediated gene therapy in past and current clinical trials. The delivery routes analyzed were direct intraparenchymal (IP), intracerebroventricular (ICV), intra-cisterna magna (CM), lumbar intrathecal (IT), and intravenous (IV). The results demonstrated that the dose used in these routes varies dramatically. The average total doses used were calculated and were 1.03 × 10^13^ for IP, 5.00 × 10^13^ for ICV, 1.26 × 10^14^ for CM, and 3.14 × 10^14^ for IT delivery. The dose for IV delivery varies by patient weight and is 1.13 × 10^15^ IV for a 10 kg infant. Ultimately, the choice of intervention must weigh the risk of an invasive surgical procedure to the toxicity and immune response associated with a high dose vector.

## 1. Introduction

The biology and function of the central nervous system (CNS) in humans is highly elaborate and currently not fully understood. Given this complexity, it is not surprising that single gene disorders often manifest with CNS disease. Disorders of CNS, including both monogenic and multifactorial disorders, comprise a significant portion of disability in both children and adults. Gene therapy for autosomal recessive genetic diseases of the CNS is conceptually attractive since the replacement of a single gene should correct or prevent disease manifestations. These single gene disorders are devastating childhood and early adulthood neurodegenerative disorders that cause poor quality of life and early mortality. Examples include aromatic L-amino acid decarboxylase (AADC) deficiency, lysosomal storage disorders such as Tay-Sachs Disease (TSD), neuronal ceroid lipofuscinoses (Batten Disease), mucopolysaccharidoses, and leukodystrophies (Canavan Disease), Huntington’s disease and neuromuscular disorders such as spinal muscular atrophy (SMA) and Duchenne muscular dystrophy (DMD). Table 1 depicts a complete list of clinical trials to date. Several of these diseases are linked to a single deficient enzyme, and the expression of that enzyme in specific cell types can potentially restore normal neurological function. There have been attempts to treat neurodegenerative diseases such as Parkinson’s and Alzheimer’s disease with gene therapy as well. Furthermore, the development of adeno-associated virus (AAV) vectors with tropism for neurons and other cells of the CNS and the ability to transduce non-dividing cells like neurons and to express therapeutic genes in very long-term fashion have made gene therapy for genetic CNS disorders feasible.

Technically accomplishing the replacement of defective genes in the CNS is, however, a daunting task, particularly since the expression of any given gene may be required in a particular CNS structure or cell type at a specific time during development. Recombinant AAV (rAAV) vectors were developed in the 1980s, initially based on AAV2, the serotype which was most extensively characterized in the pre-gene therapy era, largely because of its unique mechanisms of long-term persistence [1,2,3]. In early studies in the 1990s, AAV2 vectors proved to be capable of in vivo transduction of various tissues, including lung, muscle, and brain [4,5,6,7]. The discovery by Gao et al. of the natural biodiversity of AAV and the ability to cross-package AAV2-ITR flanked gene cassettes into AAV capsids of other serotypes led to the availability of a far wider range of rAAV vectors with a range of tissue tropisms [8,9,10]. Of the serotypes then available, the neurotropic propensity of rAAV9 and other related serotypes was discovered, along with the ability of rAAV9 vectors to cross the blood–brain barrier [11,12]. Subsequently, a wide variety of single gene disorders affecting the central nervous system have been addressed with rAAV gene therapy, with the rAAV9 vector for spinal muscular atrophy being the first one to be FDA-approved [13]. As AAV vector technology has developed, AAV vectors with different AAV capsid serotypes that have improved tropism for different cell types, including those within the CNS, have been devised [14,15,16]. However, the specific difficulty with gene or cell therapy for the CNS is the physical delivery method, particularly due to both the blood–brain barrier and the cerebrospinal fluid (CSF)–brain barrier [17,18].

AAV (adeno-associated virus) gene therapy can be delivered through various routes depending on the specific disease, target tissue, and therapeutic goals. The choice of delivery route depends on the specific disease, target tissue, safety considerations, and the desired therapeutic outcomes. Various delivery methods are continually being optimized to enhance the safety and effectiveness of AAV gene therapy. In the case of CNS gene therapy, the primary methods used have been intraparenchymal, injection into the CSF and intravenous (IV) delivery (Figure 1). Each of these will be considered in greater depth below.

Overall, adeno-associated virus (AAV) gene therapy has shown significant promise in treating various genetic and acquired diseases, but it is not without limitations and risks. Some of the potential risks and challenges associated with the delivery of AAV gene therapy include those related to innate, cellular and humoral immune responses, overexpression and off-target effects, its limited cargo capacity, tissue-specific transduction, dosage, and toxicity. In this review, we discuss alternatives to intravenous delivery and provide an overview of current clinical trials.

**Table 1 ijms-25-01050-t001:** Different delivery routes used in clinical trials, both completed and ongoing.

Disease	Specific Route	Transgene	Capsid	Dose & Volume	Ages
AADC deficiency [19,20]	Bilateral putamen [20]	hAADC	AAV2	160 µL per hemisphere: 1.81 × 10^11^ vg 200 µL per hemisphere: 2 × 10^11^ vg	1.7–8.4 years 4–19 years
	Bilateral Substantia nigra compacta and Ventral tegmental area (VTA) [19]			80 µL per hemisphere: 1.3 × 10^11^ vg–4.2 × 10^11^ vg	4–9 years
Alhzeimer’s Disease [21]	Basal forebrain (which contains the Nucleus basalis of Meynert)	NGF	AAV2	2.0 × 10^11^ (*n* = 23)	55–80 years
Batten Disease CLN2 [22] NCT00151216 NCT01161576 NCT01414985	12 intraparenchymal locations (six on each side)	hCLN2	AAV2	2.5 × 10^12^ vg × 12 = 3 × 10^12^ (*n* = 10)	3–18 years
			AAVrh.10	3 × 10^12^ particle units or 2.4–7.5 × 10^10^ vg in 150 μL × 12 = 9 × 10^11^ (*n* = 8)	2–18 years 3–18 years
Canavan Disease [23]	12 intraparenchymal locations (six on each side) frontal, periventricular, occipital	ASPA	AAV2	1.1 × 10^12^	4–83 months
Huntington’s disease [24]	Bilateral caudate	miHTT	AAV9	6 × 10^12^–6 × 10^13^	25–65 years
Metachromatic leukodystrophy (MLD) NCT01801709	12 locatinos in white matter	cuARSA	AAVrh10	1 × 10^12^; 4 × 10^12^ vg	6 months–5 years
MPS I NCT03580083	Intraparenchymal	IDUA	AAV9	1 × 10^10^–5 × 10^10^ gc/g brainmass (*n* = 5)	>4 months
MPS II NCT03566043	Intraparenchymal	IDS	AAV9	1.3 × 10^10^–2.9 × 10^11^ moi/mL per site	4 months–5 years
MPS IIIA NCT03612869	12 locations in white matter anterior, medial, and posterior to the basal ganglia [25]	hSGSH-IRES-SUMF1	AAVrh10	720 µL of 7.2 × 10^11^ vg over 12 sites (*n* = 4)	18 months to 6 years
MPS IIIB NCT03300453	16 sites in white matter	NAGLU	AAV5	4 × 10^12^ over 16 sites (*n* = 7)	18–60 months
Multiple Systems atrophy NCT04680065	Putamen	GDNF	AAV2	Dose unknown (*n* = 9)	35–75 years
Parkinson’s Disease	Bilateral postcommissural putamen CED [26,27]	hAADC	AVV2	200 µL over the four injection sites; 9 × 10^10^–3 × 10^11^ vg (*n* = 10)	57–71 years
	Unilateral subthalamic nucleus [28]	GAD	AAV2	1 × 10^12^	53–65 years
	Bilateral putamen [29,30]	hGDNF	AAV2	9 × 10^10^–3 × 10^12^ vg	>18 yo
	Bilateral putamen CED [31] Substantia nigra [32]	Neuturin	AAV2	1.3 × 10^11^–2.4 × 10^12^	
Tay-Sachs and Sandhoff Disease (GM2) NCT04669535	Bilateral thalamic convection-enhanced delivery (CED) [33]	HEXA/HEXB	AAVrh8	4.08 × 10^13^ bilaterally	6 months–12 years
	Cisterna Magna [33] and Lumbar intrathecal (IT)		AAVrh8	1 × 10^14^–4.2 × 10^13^ vg	
Krabbe Disease NCT04771416	Cisterna Magna	GALC	AAVrh10	1.4 × 10^11^–5.0 × 10^11^ gc/g brain mass (*n* = 24)	1–9 months
MPS II NCT04571970	Cisterna Magna Intracereroventricular	IDS	AAV9	Dose unkown	5–17 years
Frontotemporal Dementia (FTD) NCT04408625	Cisterna Magna	GRN	AAV9	Dose unknown	30–85 years
Canavan Disease NCT04833907	Intracerebroventricular	ASPA	AAV-oligo001	3.7 × 10^13^	3–60 months
Parkinson’s Disease NCT04127578	Cisterna Magna	GBA1	AAV9	Ascending dose	35–80 years
Gaucher Disease NCT04411654	Cisterna Magna	GBA1	AAV9	Ascending dose	0–24 months
GM1 NCT04273269	Cisterna Magna	GLB1	AAVrh10	8 × 10^12^ vg/kg	<3 years
ALS [34]	Lumbar intrathecal (IT)	miR-SOD1	AAVrh10	4.2 × 10^14^ vg	22–56 yo
CLN3 NCT03770572		CLN3	AAV9	6 × 10^13^–1.2 × 10^14^ vg	3–10 years
CLN6 NCT02725580		CLN6	AAV9	Dose unknown	>1 year
CLN7 NCT04737460		CLN7	AAV9	5 × 10^14^–1 × 10^15^ vg	1–18 years
SMA NCT03381729 [35]		SMN	scAAV9	6 × 10^13^–1.2 × 10^14^–2.4 × 10^14^ vg	6–60 months
Giant Axonal Neuropathy NCT02362438 [36]		Gigaxonin	scAAV9	3.5 × 10^13^ vg	3–99 years
Adrenomyeloneuropathy (AMN) NCT05394064		ABCD1	AAV9	1.0 × 10^15^ vg 3.0 × 10^14^ vg	18–65 years
Spinal muscular atrophy NCT03505099		SMN	scAAV9	6.0 × 10^13^ vg 1.2 × 10^14^ vg 2.4 × 10^14^ vg	6–60 months
Alzheimer’s disease NCT04133454	IV and IT	hTERT	AAV2	Dose unknown	>45 years
Krabbe NCT04693598	Intravenous (IV)	hGALC	AAVrh10	Dose unknown	<12 months
Gaucher NCT05324943		GBA1	AAVS3	Dose unknown	>18 years
Spinal muscular atrophy (SMA) [13,37]		SMN	scAAV9	1.1 × 10^14^ vg/kg	<42 days to <6 months (depending on trial)
MPS I NCT02702115		IDUA	AAV6-ZFN	Does unknown	>5 years
MPS IIIA NCT02716246 NCT04088734		hSGSH	scAAV9	0.5 × 10^13^–3.00 × 10^13^ vg/kg	2–18 years
MPS IIIB NCT03300453		hNAGLU	AAV9	2.0 × 10^13^–1.00 × 10^14^ vg/kg	18 months–60 years
GM1 NCT03952637		GLB1	AAV9	1.5 × 10^12^–4.5 × 10^13^ vg/kg	6 months–12 years
Canavan NCT04998396		ASPA	AAV9	4.50 × 10^13^ vg/kg	Up to 30 months
DMD [38] NCT05096221		micro-dystrophin	rAAVrh74	1.0 × 10^14^–3.0 × 10^14^ vg/kg	4–5 years

## 2. Routes of Delivery to the Central Nervous System

### 2.1. Intraparenchymal (IP) Delivery

Perhaps the most intuitive delivery method for AAV is directly into the site of pathology (Figure 1). This method is effective in the treatment of diseases that affect a very specific anatomic location, such as AADC deficiency, in which the gene function is only necessary within the basal ganglia. The first ever intraparenchymal clinical trial using AAV was in Canavan Disease, where larger brain areas were targeted utilizing twelve sites of injection [6,23]. Other studies in Batten Disease have also used this multiple injection approach [22,39]. More recently, direct brain injection into the thalamus has been used to take advantage of the existing axonal tracts within the brain for retrograde and anterograde transport [40,41,42,43,44]. Interestingly, these early-generation therapies for AADC deficiency, Canavan Disease and Batten Disease utilized the AAV2 capsid, which has relatively low tropism for CNS tissues. Upstaza, the current EMA-approved agent for AADC deficiency, utilizes an AAV2 capsid. Second-generation vectors, such as AAV9, are considerably more efficient at CNS transduction, bypassing the blood–brain barrier and showing potential for the treatment of global CNS pathology [45]. This was highlighted by the FDA approval of intravenously administered AAV9 for spinal muscular atrophy.

Several years of work with AAVs has demonstrated reliable efficacy in delivering genes to specific areas of the CNS with direct intraparenchymal injection via stereotactic guidance [46,47]. The benefit of this route is that it bypasses the need for the capsid to cross the blood–brain barrier and allows for smaller doses of vector. In the clinical trials that have been completed or are ongoing, the average doses that were used were 1.03 × 10^13^ for intraparenchymal delivery, 5.00 × 10^13^ for intracerebroventricular delivery, 1.26 × 10^14^ for cisterna magna delivery, and 3.14 × 10^14^ for lumbar intrathecal delivery (Figure 2). The dose for IV delivery varies by patient weight and is 1.13 × 10^15^ IV for a 10 kg infant. Stereotactic brain delivery of genes has targeted the striatum (putamen), thalamus, substantia nigra, ventral tegmental area, nucleus basalis of Meynert, white matter in the frontal, periventricular and occipital lobes. The safety of intracranial AAV-mediated gene delivery was first shown in patients with Canavan disease, a lethal leukodystrophy. Thereafter, several studies on Parkinson’s disease utilized AAV2 to target different pathways of dopamine production, such as glutamic acid decarboxylase (GAD) into the subthalamic nucleus, aromatic L-amino acid decarboxylase (AADC) into the putamen and, also genes linked to neuronal growth such as GDNF and neurterin.

Success in the field was highlighted recently by the EMA approval of eladocagene exuparvovec, Upstaza, for aromatic L-amino acid decarboxylase (AADC) deficiency, a fatal error of neurotransmitter biosynthesis resulting from mutations in the dopa decarboxylase (DDC) gene. The one-time treatment corrects the underlying genetic defect by using AAV2 to deliver a functioning DDC gene directly into the putamen. The authors chose to inject the putamen because the putamen receives dopaminergic projections from the substantia nigra, and dopamine depletion in the putamen causes the loss of voluntary motor movement [20]. This expression of the AADC enzyme and restoration of dopamine resulted in improved motor function in patients [48,49]. A one-time single injection of a total dose of 1.8 × 10^11^ vg delivered as four 0.08 mL (0.45 × 10^11^ vg) infusions directly to four quadrants of the putamen was performed. After the infusion of Upstaza into the putamen, the AADC enzyme is thought to be expressed through the direct transduction of postsynaptic neurons in the putamen that is thought to synthesize dopamine.

Another group led by Bankiewicz et al. published a different target for gene therapy for AADC deficiency; the substantia nigra compacta (SNc) in the midbrain and ventral tegmental area (VTA). Both the SNc and the VTA are the highways for dopamine production in the brain. By delivering AAV2-hAADC to the SNc and VTA, AADC enzyme activity is directly increased in the midbrain dopaminergic neurons, resulting in dopamine synthesis and rescue of neurotransmission through the nigrostriatal, mesolimbic, and mesocortical pathways. Midbrain delivery also takes advantage of anterograde axonal transport of AAV2 from these regions to deliver AAV2-hAADC to neuroanatomically appropriate brain regions like the striatum [19].

The idea of treating the entire brain with one or two injection sites is ambitious. Multiple intracranial injections to the white matter were studied in Batten disease and mucopolysarcharidosis (MPS, mainly IIIa and IIIb); however, these results, although generally showing safety, were not ultimately successful in clinical treatment. This could be a result of AAV2 utilization initially, and AAV9 is currently being investigated for the same diseases utilizing the intrathecal route.

Another intracerebral route that may have some promise is bilateral intrathalamic delivery, and this is due to the fact that the thalamus is a relay station for the entire cerebral cortex. The thalamus is an elaborate structure consisting of several nuclei that connect to almost all areas of the cortex, and these pathways can serve as highways for axonal transport [50]. Preclinical animal models have shown that a single injection of AAV carrying Gm1 into the thalamus can cause widespread distribution of this enzyme through the entire injected brain hemisphere [51]. Bilateral thalamic injections for gene replacement in GM2 mice, cats, and sheep were carried out by Gray-Edwards and Esteves et al., and this comprehensive work ultimately led to the clinical trial in GM2 patients utilizing bilateral thalamic injections. No adverse events have been reported in the small number of patients treated with this method [33].

For a complete list of past and current clinical trials involving intraparenchymal delivery, please see Table 1.

#### 2.1.1. Convection-Enhanced Delivery

Convection-enhanced delivery (CED) using neurosurgical techniques has been researched predominantly by Bankiewicz et al. [46]. CED uses a hydrostatic pressure gradient, also known as bulk flow, to maximize the spread of compounds throughout the brain. A syringe pump is used to infuse small amounts of volume (approximately 1–4 μL/min) through a specifically designed (usually silica) stepped cannula into a specific brain region. A stepped cannula is used to prevent fluid backflow. With CED, high volumes of vector can be delivered through interstitial spaces in a way that minimizes potential backflow and brain cavitation. Cranial navigation software and hardware such as Brainlab^®^ and ROSA^®^ can accurately plan trajectories and allow for more precise catheter placement. At UMass Chan Medical School and UMass Memorial Health, the authors use the Brainlab Flexible Catheter [REF19772] and ROSA One software version 3.1.6 for intraparenchymal convection enhanced gene delivery navigation guidance. Although stereotactic placement of a cannula in a deep brain region for convection-enhanced delivery requires neurosurgical expertise, there are several advantages to this technique of drug delivery. For instance, it is possible to target a specific set of cells in the brain, such as the medium spiny neurons and/or monoenzymatic/dienzymatic neurons, to enhance dopamine production in the putamen. It is also possible to inject in the ventral tegmental area or substantia nigra, again for dopamine production. In this way, treatment can be targeted to diseased neuronal circuitry and can avoid normal brain pathways.

The use of intraoperative imaging in AAV vector delivery is helpful for accurate targeting and delivery. Intraoperative CT and MRI can confirm the accurate placement of the cannula tip. Co-infusion of the AAV with a surrogate imaging tracer (gadoteridol) has also been used and has reportedly been shown to be safe, accurate and reliable in determining the infusate dynamics [52,53]. Intraoperative MRI during the infusion allows for real-time tracking of the infusate, allowing for the identification of initial infusion, the identification of backflow, and the realization of perfusion outside of the target. If a problem is detected during infusion, adjustments to the catheter tip can be made intraoperatively before the infusion is complete [54].

#### 2.1.2. Axonal Transport

Axonal transport is a physiological process used to transport material in between the cell body and the axonal terminal of neurons. Depending on the direction of flow, whether to the axonal terminal or to the cell body, axonal transport is divided into anterograde and retrograde axonal transport, respectively [55]. The axonal fiber tracts of the human brain have been studied for years, and areas connected by axonal fibers are well characterized. Manipulation of these interconnected sites for therapeutic gain can be influenced by the target selection and AAV serotype. Specifically, the perfusion of a single anatomical target, like the ventral tegmental area or the thalamus with AAV can be used for distant widespread transgene expression that is defined by anterograde and/or retrograde transport depending on serotype [43]. AAV vectors can take advantage of these axonal highways to deliver transgenes both antero- and/or retrograde from the site of delivery. Previous work has demonstrated that AAV delivered by CED can be transported to interconnected regions by antero- or retrograde axonal transport that is serotype-dependent [40,56,57,58]. While AAV2 undergoes anterograde transport and AAV6 undergoes retrograde transport [40], studies have demonstrated that AAV serotypes 1, 5, 8, and 9 undergo both anterograde and retrograde transport [46,56,59].

#### 2.1.3. Cross Correction

Convection-enhanced delivery allows for the delivery of high payloads of vector to specific areas of the brain, and axonal transport then mediates delivery to connected neurons. However, these alone are not sufficient for the transduction of the whole brain. As discussed previously, AAV serotypes guide cell-specific transduction, and the brain consists of many cell types, including neurons, microglia and astrocytes. A natural aspect of the biology of lysosomal enzymes has been harnessed to treat lysosomal storage diseases (LSDs). When one neuron is transduced, it can either transport its transgenic protein across to another neuron via axonal transport, or in the case of a secreted protein, it can secrete the protein in the extracellular matrix, and nearby cells can be cross-corrected via the mannose-6-phosphate receptor. This process, which utilizes the mannose-6-phosphate receptor-mediated endocytic pathway, is called cross-correction. Cross-correction is an important aspect of gene therapy when it pertains to lysosomal storage diseases, where the missing enzyme can be produced in a few transduced cells but can then be released and uptaken by surrounding cells as first described in the case of mucopolysaccharidosis type 1 [43,60]. The concept of cross-correction is utilized in the treatment of mucopolysachharidosis, metachromatic leukodystrophy and GM2 diseases such as Tay-sachs and Sandhoff Disease, where even a minimal re-establishment of enzymatic activity can have dramatic clinical implications [61].

#### 2.1.4. Toxicity Associated with Direct Intraparenchymal Injection

Gene therapy utilizing AAV delivered to the brain parenchyma has been generally well tolerated surgically with few adverse events. Although rare, adverse effects related to the neurosurgical procedure include misplaced cannula for vector infusion, infusate backflow, infusate leaking into surrounding structures, hemorrhage, seizure, and wound problems. A notable case of focal lesions in the brain following intracerebral gene therapy for mucopolysaccharidosis IIIA was reported by Bugiani et al. in 2023 [62]. In this trial, called the AAVance trial, patients were immunosuppressed 7 days prior to surgery and up to 1 year post-surgery with mycophenolate, tacrolimus and prednisolone. They reported that three months after stereotactic infusion of AAVrh.10-SGSH, a patient developed radiographic lesions surrounding the injection sites on MRI, and one of these lesions caused temporary neurological deficits. Over time, the lesions stabilized and decreased in size. A brain biopsy of one of the lesions showed no abundance of B or T cells, strong expression of transgene (sulfamidase), and no detectable heparan sulfate. The authors interpreted these results as an overexpression of sulfamidase locally at the site of gene therapy, which caused dysfunction of transduced cells and extracellular spilling of lysosomal enzymes [62]. Another case of intracranial pathology was observed in an earlier clinical trial for metachromatic leukodystrophy (MLD), a lysosomal disease caused by a defect in the arylsulfatase A (ARSA) gene. In this trial, the same capsid as in the aforementioned trial, AAVrh.10, was used to deliver hARSA to the white matter of the centrum semiovale of four children. All patients received corticosteroids. One serious adverse event was reported, and it was called “intracranial suffusion”, which reportedly resolved spontaneously [63]. In an even earlier trial involving a form of Batten Disease, late infantile neuronal ceroid lipofuscinosis (LINCL), patients received AAV2 expressing the human CLN2 cDNA in 12 frontal and parietal white matter locations [39]. One patient developed a status epilepticus on day 14, which was ultimately fatal. It was unclear if this was related to the treatment or to the natural history of LINCL [39].

More recently, and also thought to be related to the overexpression of transgene, patients in the AADC clinical trials for the recently EMA-approved medication, Upstaza, experienced dyskinesia after treatment. This was temporary and resolved over time [20].

#### 2.1.5. Immune Response Associated with Direct Intraparenchymal Injection

While the CNS has been described as a relatively immune-privileged site, direct intraparenchymal injection of AAV vectors may result in significant systemic biodistribution, depending on the dose, accompanied by systemic immune responses. While known interactions of AAV with toll receptors and other pattern recognition receptors would lead one to predict localized innate immune responses as well, such responses have generally been mild or even subclinical.

However, adaptive immune responses to AAV after intraparenchymal injection of AAV have been documented [33,64]. The development of anti-AAV antibodies after vector delivery, while observed in several studies, is not necessarily problematic since vector redosing is generally not necessary given the long-term persistence of AAV vector expression in non-dividing cells of the CNS. The presence of effector T cell responses against AAV capsids has been observed as well. These may be accompanied by modest elevations of liver enzymes and are generally treatable with systemic corticosteroids [65,66,67,68].

### 2.2. Intravenous (IV) Delivery

The intravenous delivery (IV) delivery of AAV to cross the BBB was not thought to be feasible prior to the discovery that the AAV9-based vectors possessed some ability to cross the BBB [12]. AAV9 vector efficiency at penetrating the BBB has proven to be sufficient for the delivery of the SMN1 gene to lower motor neurons in spinal muscular atrophy type 1, leading to the FDA approval of the AAV9-SMN1 vector, Zolgensma [13]. Although delivery to the CNS directly seems intuitive, systemic delivery may be more effective at widespread transduction of the CNS and other peripheral organs. This has been exemplified in the groundbreaking clinical trial for spinal muscular atrophy (SMA), where Onasemnogene abeparvovec, also known as Zolgensma, is delivered IV and crosses the blood–brain barrier. Zolgensma is a gene therapy that was recently approved in 2019 by the US Food and Drug Administration as a treatment for SMA in pediatric patients (<2 years old). Zolgensma consists of a single-dose, intravenous infusion of a self-complementary adeno-associated vector 9 (AAV9) that crosses the blood–brain barrier and delivers a functional copy of the SMN1 gene under the control of the cytomegalovirus (CMV) enhancer/chicken-β-actin-hybrid promoter (CB).

Although intravenous AAV9 is efficacious for gene transfer, the doses required for delivery to the central nervous system and for bypassing the central nervous system are very high, at or exceeding 1 × 10^14^ vg/kg. This is because the vast majority of IV AAV genomes are distributed to the liver and other organs rather than to the CNS. The major biodistribution of AAV to organs outside the CNS leads, in a small subset of patients, to clinically significant hepatotoxicity, thrombocytopenia with or without thrombotic microangiopathy (TMA) or cardiac toxicity [68,69,70,71,72,73,74]. In addition to these considerations, IV AAV gene therapy may be inhibited by pre-existing neutralizing antibodies which, therefore, may exclude certain patients from being eligible to receive the therapy. Zolgensma highlighted the potential of AAV-mediated gene therapy to cross the blood–brain barrier and paved the way for other clinical trials. Unfortunately, high-dose systemic therapy (i.e., 3 × 10^14^ GC/kg) for neuromuscular diseases like X-linked myotubular myopathy [75] or early clinical trials for SMA have proven that very high doses can cause hepatotoxicity and death. Even with Zolgensma, which was approved in 2019, data revealed that there are dose-limiting toxicities, including hepatotoxicity [72].

Canavan Disease (CD) is a leukodystrophy caused by mutations in aspartate acylase (ASPA), which cause excessive buildup of N-acetyl aspartic acid in the brain parenchyma. This leads to brain edema and abnormal myelination. CD was an example of AAV application in the human brain. Although initially thought to be best treated with direct injection into several regions of white matter parenchyma, CD has been recently successfully treated using the IV route alone (CANaspire Trial, NCT04998396). The first proof-of-concept intraparenchymal delivery of AAV2-ASPA performed for CD was in two children by Leone et al. in 2002 but it was not until 2012 when a follow-up study demonstrated some success in disease stabilization and slowed progression of brain atrophy and seizures [23]. Importantly, there were no adverse events associated with the direct brain injections of vector, and this paved the way for future intracerebral injections for gene therapy. Subsequent preclinical data utilizing a mouse model of CD showed that ASPA gene replacement provided phenotypic rescue when using a capsid that is systemically delivered and crosses the blood–brain barrier: AAV9 [76]. Currently, a clinical trial is investigating the systemic delivery of AAV9-ASPA under the control of a ubiquitous promoter to restore ASPA expression in both neuronal and non-neuronal cell types (NCT04998396). To better understand whether a dual delivery method enhances the efficacy of AAV9-CB6-ASPA, a 2-year-old child with Canavan disease was treated with IV and ICV delivery in an expanded access trial. The results showed that the child had remyelination in the CNS, restoration of visual function, neurodevelopmental improvement and reduction in N-acetytalaspartate (NAA) accumulation at 4 years after dosing. The authors concluded that the combination of dual routes of therapy with immune suppression can improve therapeutic outcomes by increasing vector distribution [77].

#### 2.2.1. Biodistribution of AAV Delivery

In 2023, Gray et al. [78] published a comprehensive review of the biodistribution of AAV gene therapy in the CNS. They analyzed the preclinical studies of AAV gene therapy biodistribution following cerebrospinal fluid delivery (intracerebroventricular, intra-cisterna magna, and intrathecal lumbar). They concluded that the current preclinical literature varies greatly in the reported biodistribution of AAV following administration in the CSF. This variability, the authors postulated, was due to differences in the animal model used, vector serotype used, method used to detect biodistribution, route of administration, and dose. In terms of detecting biodistribution, methods for its evaluation include vector copy number and should also report details of the empty:full capsid ratio and quality of the encapsidated genome.

#### 2.2.2. Immune Response Associated with IV Delivery of AAV

The immune response to adeno-associated virus (AAV) vectors can have significant implications for the success of gene therapy and other AAV-based treatments. AAV vectors are derived from a benign virus, but they can still trigger an immune response. The immune response to AAV can be categorized into two main components: the innate immune response and the adaptive immune response. The innate immune response is comprised of the inflammatory response and complement activation. Upon administration of AAV vectors, the body’s innate immune system may recognize the foreign viral particles via interactions of AAV capsid and AAV vector DNA with TLR9 and other pattern recognition receptors and may respond with an inflammatory reaction. This can involve the release of proinflammatory cytokines and activation of immune cells like macrophages and dendritic cells. One strategy that has been used to reduce such toxicity is the elimination of CpG sequences within the vector genome [79].

The complement system may be a part of the innate immune response and may become activated in response to AAV vectors. This can lead to the elimination of AAV particles and to other systemic toxicities, including thrombocytopenia and endothelial injury. For these reasons, it is important to assess the patient’s baseline neutralizing antibody levels prior to enrollment in gene therapy clinical trials.

The adaptive immune system can generate either antibodies or cytotoxic T cell responses, which may be directed against AAV capsid proteins and transgene proteins. These antibodies can neutralize AAV vectors, preventing them from effectively delivering therapeutic genes to target cells. AAV-specific T cells may recognize and target AAV-infected cells. This T-cell response can be both cellular (CD8+ cytotoxic T cells) and humoral (CD4+ helper T cells). In some cases, a strong T-cell response can result in the clearance of AAV-transduced cells [67,71].

Tissue macrophages of the liver, known as Kupffer cells, can present the new transgene protein to either B or T cells. The therapeutic effect of the desired transgene protein may be eliminated by either developing antibodies to the transgene protein or by using cytotoxic T cells to remove the transduced cells [80,81]. Anti-capsid cytotoxic T-cell-mediated destruction of transduced hepatocytes in the clinical setting may be mitigated with prophylactic or on-demand immunosuppression [67].

#### 2.2.3. Clinical Toxicity Associated with IV Delivery of AAV

Preclinical studies have suggested that cell-mediated immunity directed against the AAV capsid plays an important role in the safety and efficacy of AAV gene transfer in humans. Immune responses have been observed in AAV vector clinical trials across different neuromuscular diseases (e.g., SMA, Duchenne muscular dystrophy, myotubular myopathy).

The intravenous delivery of AAV vectors carries the risk of hepatotoxicity. This may be represented by elevation of liver enzymes (alanine aminotransferase, aspartate aminotransferase or γ-glutamyl transferase). Thus far, hepatotoxicity has resulted in the tragic death of eight patients treated with AAV gene therapy, including four patients treated for X-linked myotubular myopathy and four patients treated for spinal muscular atrophy.

X-linked myotubular myopathy is a defect in the *MTM1* gene that causes skeletal muscle weakness in 80% of affected boys, and without intervention, greater than 50% of these children will die within the first 18 months of life. The ASPIRO trial enrolled 24 boys with X-linked myotubular myopathy who were on mechanical ventilatory support for treatment with an AAV8 delivering the human *MTM1* transgene (1.3–3 × 10^14^ vg/kg). Unfortunately, four of these patients died, and all deaths were associated with severe cholestatic liver injury [82,83]. The relationship of the toxicity to vector administration remains unclear but appears to relate to pre-existing hepatobiliary disease in these patients [84].

Clinical trials for spinal muscular atrophy (SMA) have demonstrated clear evidence of clinically meaningful efficacy following IV administration of onasemnogeneab eparvovec (Zolgensma—1.1 × 10^14^ vg/kg) in spinal muscular atrophy; however, there were also four deaths reported in the 5–6-week post-marketing stage [85]. The FDA has also issued a black box warning for serious liver injury and acute liver failure for its use and recommended a 30-day course of prophylactic prednisolone beginning just prior to treatment (https://www.fda.gov/media/126109/download, accessed on 30 November 2023). Chand et al. in the *Journal of Hepatology* analyzed 325 patients with SMA in five clinical trials and found that 90 of 100 patients had elevated alanine aminotransferase, and or aspartate aminotransferase and or bilirubin concentrations. Of these, 34% had liver-associated adverse events, and two patients had serious acute liver injury. All of the patients were treated with prednisolone for 60–120 days, and the liver injury resolved [68]. In addition, cardiac toxicity, manifested by high troponin levels, has also been noted with this vector.

Two new AAV-related immunotoxicities emerged in the last few years. These included dorsal route ganglia toxicity and a syndrome of thrombocytopenia, hepatic and renal toxicity, referred to as thrombotic microangiopathy [86]. Dorsal root ganglia toxicity is discussed below, specifically in relation to intrathecal delivery. Thrombotic microangiopathy (TMA) has been reported in a few cases treated with Zolgensma [73,74]. TMA, with complement activation, has also been reported in cases of AAV9-based gene therapy for Duchenne muscular dystrophy and Danon disease [74]. The patient with Danon disease developed TMA, but this resolved with supportive treatment, including transient hemodialysis [87]. It should also be noted that thrombocytopenia, or at least a drop from baseline platelet counts, has been seen frequently, including in patients where there is no evidence of complement activation or other aspects of TMA.

Acute lung injury, which presented as capillary leak syndrome, was also observed in one case of Duchene Muscular Dystrophy (DMD) reported in NEJM after the systemic administration of an AAV9 vector expressing VP64 (transcription activation domain)-fused dead Staphylococcus aureus Cas9 (dCas9-VP64) from the muscle-specific CK8e promoter, and a gRNA targeting the cortical promoter of the DMD gene [88].

### 2.3. Delivery to the Cerebrospinal Fluid (CSF)

Stereotactic intracerebroventricular (ICV), intracisternal magna (ICM) and intrathecal (IT) injections to deliver AAV via the cerebrospinal fluid (CSF) for widespread distribution to the CNS are currently being investigated in clinical trials. Although these approaches may be advantageous for diseases with widespread CNS pathology, the CSF-brain barrier is still a challenge for parenchymal transduction. The presence of tight junctions between ependymal cells in the ventricle is a significant barrier for some AAV serotypes, which must pass through these cells from the ventricle to have intraparenchymal spread. Interestingly, Chakrabarty et al. found that not only are there differences in capsid serotype transduction of the CNS, but there are cell-type differences (neurons versus glial cell) in the transduction profile of an ICV injection carried out in mice at postnatal day 0 (p0) when compared to post natal day 1 or 2 (P1–P2) [89]. Whether this is due to structural changes in the postnatal brain or differential expression of receptors on the surface of cells depending on age is largely unknown.

One might assume that an injection into the human CSF, whether ICV, ICM or IT, yields the same transduction profile. Historically, the flow of CSF has been known to be unidirectional: CSF produced by choroid plexus mainly in the lateral ventricles travels into the third ventricle, through the cerebral aqueduct, into the foramina of Magendie and Luschka into the cerebral cisterns and subarachnoid spaces or out the central canal and into the lumbar intrathecal space, and this flow is largely mediated by cilia in the ependymal lining of ventricles. This concept has been challenged by several scientists who claim that CSF flow is through the brain interstitium itself and the so-called “glymphatic system”. Nedergaard et al. showed via in vivo two-photon imaging of small fluorescent tracers that CSF enters the brain parenchyma along paravascular spaces that surround penetrating arteries and that brain interstitial fluid is cleared along paravenous drainage pathways. They also showed that ventricular CSF minimally enters the brain parenchyma whereas subarachnoid CSF rapidly enters the brain parenchyma [90,91]. It is, therefore, probable that the delivery of gene therapy into the CSF, whether by the ventricle, cisterna magna or the lumbar intrathecal space, may yield differing results.

Whether there is a difference in transduction when an AAV capsid containing transgene is administered via ICV, ICM or IT was studied by Hinderer et al. In 2014, they showed that an AAV9 injection into the cisterna magna of a non-human primate (NHP) was 100-fold more efficient at transducing the brain and a 10-fold more efficient at transducing the spinal cord when compared to lumbar intrathecal delivery [92]. In a different study, Hinderer et al. demonstrated that an AAV9-GFP injection (1.8 × 10^13^ in 1 mL) into the cisterna magna could effectively target the entire brain, as demonstrated by widespread GFP staining but predominantly the cerebellum, brainstem and spinal cord. Similarly, a unilateral injection in the ventricle (*n* = 3) causes widespread transduction of the bilateral brain, however, a cisternal injection (*n* = 3) has greater transduction of the spinal cord [93]. The team noted that one of the dogs did not survive the ICV injection and that this was due to a T cell-mediated immune response to the transgene (GFP) at the site of injection. Additional histological evaluation revealed that after ICV injection, there are areas of severe lymphocytic inflammation in the region of the injection site, whereas the ICM injection did not have this associated toxicity due to the fact that one does not need to trespass parenchyma to deliver to the cisterna magna. Altogether, this preclinical evidence suggests that ICM and ICV routes of delivery are both promising alternatives to direct parenchymal injection for broad CNS transduction. There are concerns regarding the risks of direct cisterna magna puncture in humans due to the proximity of critical structures in this area, such as the brainstem and upper cervical cord. To overcome this concern, the cisterna magna can be accessed via a lumbar access site using an endovascular SL-10 catheter to advance the catheter to the cisterna magna, as demonstrated by Taghian et al. [94].

#### 2.3.1. Lumbar Intrathecal (IT) Delivery

Intrathecal delivery involves delivering the AAV vector directly into the cerebrospinal fluid (CSF) through a lumbar puncture. Intrathecal delivery enables the AAV to target the central nervous system (CNS) and treat neurological disorders, such as spinal muscular atrophy or certain types of lysosomal storage diseases.

Several current clinical trials use intrathecal routes of delivery for CNS penetration. These include, Alzheimer’s disease (NCT03634007), Giant axonal neuropathy (NCT02362438), Batten Disease [CLN3—NCT0377052, CLN6—NCT02725580, CLN7—NCT04737460], GM2 (NCT04669535), SMA2 (NCT03381729 & NCT05089656) and IGHMBP2-related diseases (NCT05152823).

The challenge with these methods of delivery is again widespread delivery to the entire CNS, which is limited by the CSF brain barrier.

#### 2.3.2. Intracerebroventricular (ICV) Delivery

Access to the cerebral ventricles, which can be performed with or without stereotactic guidance, is a relatively common neurosurgical procedure. Clinical trials currently investigating its use are those in Canavan disease (NCT04833907) and MPS II (NCT04571970).

#### 2.3.3. Cisterna Magna (CM) Delivery

The cisterna magna is a large chamber or cistern that bathes the cerebellum and brainstem. It has historically been used as a method for delivery to the spinal fluid in animals, but due to the proximity of the cisterna magna to vital structures such as the brainstem, it is not the preferred method of delivery in humans. Taghian et al. reported the development of a gene therapy delivery method to the cisterna magna through the adaptation of an intravascular microcatheter, which can be safely navigated intrathecally under fluoroscopic guidance [94]. The team reported the safety, reproducibility, and distribution/transduction of this method in sheep using a scAAV9-GFP vector. This technique was used to treat two Tay-Sachs disease patients with AAV gene therapy [33,94]. No adverse effects were observed during infusion or post-treatment. This delivery technique is a safe and minimally invasive alternative to direct infusion into the cisterna magna, achieving broad distribution of AAV gene transfer to mainly the cerebellum and spinal cord but also distant cortical areas.

Current clinical trials that are using an ICM delivery route include Frontotemporal Dementia-GRN (NCT04408625 and NCT04408625), Parkinson’s disease-GBA1 (NCT04127578), Gaucher disease (NCT04411654), GM1 (NCT04713475), GM2 (NCT04669535), Krabbe disease (NCT04771416), MPS I (NCT03580083), and MPS II (NCT03566043).

#### 2.3.4. Toxicity

The main toxicity associated with delivery to the CSF or intrathecal delivery is dorsal root ganglia (DRG) toxicity. Although only described once in a human [34], dorsal root ganglia toxicity has been repeatedly found in animal models [93]. Wilson et al. studied the presence and degree of DRG pathology in animals treated with a wide variety of capsids. They found that all five capsids and all five promoters tested had some degree of DRG pathology in non-human primates (NHPs). Additionally, there was DRG pathology associated with 20 different transgenes that were tested. They showed that DRG pathology is almost universal in all NHPs tested. The team postulated that this is because of toxicity mediated by transgene overexpression. However, the pathologic changes were microscopic, and the animals did not display clinical signs. The incorporation of sensitive techniques such as nerve conduction velocity testing was, however, able to pick up peripheral sensory neuropathy.

The first in-human case of possible DRG toxicity was reported by Brown et al. after using an intrathecal delivery of AAV containing a microRNA to suppress SOD1, the pathologic gene in familial amyotrophic lateral sclerosis (ALS). Three weeks after the infusion, the patient had transient tingling in both hands, and 1 week later, he reported having a feeling of painful “electric shocks” in his left foot. Cerebrospinal fluid analysis revealed pleocytosis with high protein (23 white cells and a protein level of 342 mg/dL). Simultaneously, the patient had an elevation in hepatic aminotransferases. EMG analysis at 10 weeks revealed absent or reduced sensory nerve potentials, and MRI at 16 weeks post-infusion showed enhancement in the cauda equina and some DRGs. High-dose steroids were used for treatment, and over the course of several weeks, the pain in the patient’s left foot lessened [34].

## 3. Summary and Analysis

Gene therapy utilizing viral vectors continues to hold promise for the treatment of neurological disorders, as evidenced by the U.S. Food and Drug Administration (FDA) or European Medical Agency (EMA) approvals of AAV-mediated gene therapy for Luxturna (FDA), Zolgensma (FDA), and Upstaza (EMA). As viral vectors continue to be modified and optimized for delivery to the cells of the central nervous system, it is important to strictly uphold parameters for safety and efficacy.

We presented here an analysis of the delivery methods for AAV-mediated gene therapy in completed and ongoing clinical trials. The delivery routes analyzed were direct intraparenchymal (IP), intracerebroventricular (ICV), intra-cisterna magna (CM), lumbar intrathecal (IT), and intravenous (IV). The results demonstrated that the dose used in these routes varies dramatically. The average total doses used were calculated and were 1.03 × 10^13^ for IP, 5.00 × 10^13^ for ICV, 1.26 × 10^14^ for CM, and 3.14 × 10^14^ for IT delivery (Figure 2). The dose for IV delivery varies by patient weight and is 1.13 × 10^15^ IV for a 10 kg infant. Toxicities associated with IP delivery have included vector overexpression in the brain and seizures. A concerning form of toxicity associated with delivery to the CSF (IT or CM), although more commonly observed in animals rather than humans, is dorsal root ganglia toxicity. All forms of delivery can be associated with systemic toxicity, as seen most prominently in the liver, especially with IV delivery. Additionally, intravenous delivery has been associated with thrombocytopenia and thrombotic microangiopathy. All forms of administration of AAV have been linked to a host immune response, whether by the innate immune system, antibody-mediated response, or, more commonly, an adaptive T cell response to the viral capsid. It remains important to screen patients for neutralizing antibodies to the viral capsid and to prophylactically immunosuppress patients utilizing both corticosteroids and B and T cell depletion medications prior to gene therapy.

As with any other therapy, AAV gene therapy for CNS disorders requires a balance of safety and efficacy concerns. The specific procedures associated with each route of delivery vary in their degree of risk, with the more direct and invasive procedures incurring more procedural risk than IV administration. Conversely, the dose of vector required to achieve therapeutic effect is much higher with IV administration, which has the potential to increase systemic exposure and immune-related risks. Given this tradeoff, there is no single best approach at the current time. Regardless of the route of delivery, the risk/benefit relationship will be improved if vector potency can be improved, thereby allowing for greater transgene expression for any given dose of vector. We expect that as vector platforms continue to improve, this will enhance the risk-benefit balance for patients who are in need of these potentially life-sustaining therapies.

## 4. Literature Review Methods

This review consisted of a literature search on Pubmed and a ClinicalTrials.gov search. Keywords for both searches consisted of “adeno-associated virus” and “central nervous system”. When possible, information was obtained from published literature. If the published literature was not available, information was obtained from clinicaltrials.gov.

Diseases that have been studied utilizing gene therapy in clinical trials include aromatic L-amino acid decarboxylase (AADC) Deficiency, Spinal Muscular Atrophy (SMA), Alzheimer’s Disease, Parkinson’s Disease, Batten Disease or Neuronal Ceroid Lipofuscinosis (CLN2, CLN3, CLN6, CLN7), Canavan Disease, Huntington Disease, Metachromatic Leukodystrophy (MLD), Frontotemporal Dementia (FTD), Mucopolysaccharidosis Type I (MPS 1), Mucopolysaccharidosis Type II (MPSII), Mucopolysaccharidosis Type IIIa (MPSIIIa), Mucopolysaccharidosis Type IIIa (MPSIIIb), Multiple Systems Atrophy (MSA), Tay-sachs and Sandhoff Diseases or GM2 gangliosidosis, Krabbe Disease, Amyotrophic Lateral Sclerosis (ALS), and Duchene Muscular Dystrophy (DMD. These are further elucidated in Table 1.

## Figures and Tables

**Figure 1 ijms-25-01050-f001:**
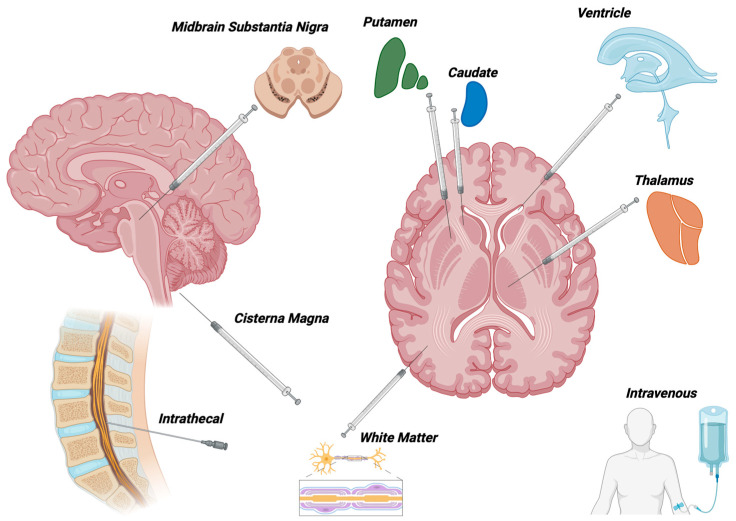
Modes of delivery for AAV-mediated gene therapy to the nervous system used in clinical trials. Direct needle trajectory is not represented anatomically and varies by patient. Created utilizing Biorender.

**Figure 2 ijms-25-01050-f002:**
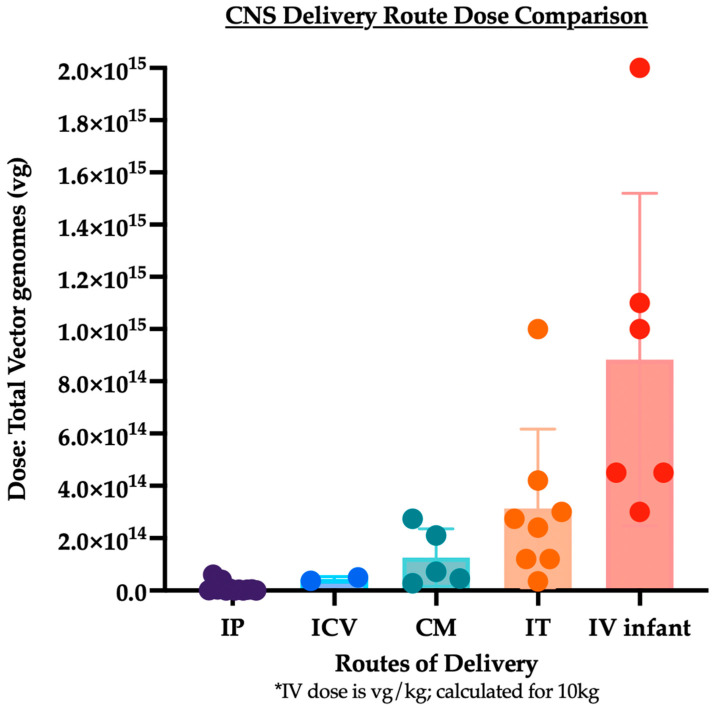
Bar graph depicting average doses used in various delivery routes for AAV-mediated gene therapy to the central nervous system. IP = intraparenchymal; ICV = intracerebroventricular; CM = intra-cisterna magna; IT = lumbar intrathecal; IV = intravenous. For IP, ICV, CM and IT, the dose is a constant vector genome amount (vg). * For IV dosing, the dose is vg/kg. The dose in this graph demonstrates the total dose for a 10 kg infant. Adult IV dosing is much higher. For more information, see Table 1.

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
