# Peer review of "Delivery of Adeno-Associated Virus Vectors to the Central Nervous System for Correction of Single Gene Disorders"

_ijms, 2024, doi:10.3390/ijms25021050_

Round 1
Reviewer 1 Report
Comments and Suggestions for Authors
Authors in this manuscript review an important field of study wherein viral vectors (i.e., AAV) are used to deliver genes to the CNS for correcting single gene disorders. This is an important and hot area of work therefore this is a timely review in that regard.
I would have following suggestions to further improve the quality/readability of this review.
1. The introduction starts on a good note however there is missing introduction to the diseases e.g., single gene disorders. It would be helpful to list and discuss them so that the reader has an understanding of the same. There is a table at the end of the manuscript however a discussion and reference to the same should be done in the introduction section as well.
2. Another missing link is an introduction and discussion about AAV vector itself including a figure of the same, also including various types of AAV vectors, their pros and cons with respect to their usage for CNS single gene disorders.
3. Summary and analysis section (section 2.4) is also short and does not go beyond the superficial summary. More discussion should be added to the section moreover this can be made into it's own section (e.g., section 3: summary and analysis). Tying it to 'section 2: Routes of Delivery to the Central Nervous System' does not make complete sense, at least to me.
Reviewer 2 Report
Comments and Suggestions for Authors
Title: Delivery of Adeno-Associated Virus Vectors to the Central Nervous System for Correction of Single Gene Disorders.
Authors: Daci, R., Flotte, T. R.
Summary: This paper provides a nice review of the current status of using AAV as a gene therapy vector to correct or treat diseases of the central nervous system (CNS). Different routes of delivery with the challenges and advantages of each are presented, as it relates to both efficacy and safety. Because IV delivery requires larger doses of vector, this has been shown to have a reduced safety profile, so alternatives to IV delivery are presented, as well as an overview of recent clinical trials exploring these alternatives. Different AAV vectors with different AAV capsid serotypes with improved tropism have been developed to achieve gene delivery to specific regions of the CNS (ie. AAV2, AAV5, AAV6, AAV9, AAVrh8, AAVrh10, AAVrh74, AAVS3) and for a host of CNS disorders. Data for these vectors is described, including targeting to specific areas of the CNS, the delivery method, efficacy and obstacles presented by the immune response. This paper will be of interest to the readers of this journal.
Revisions needed:
Page 4 Line 107 – change the word resulting to resulted.
Page 4 Line132 – delete the letter t after into.
Page 7 Line 283 – need a citation after the word alone.
Page 9 Line 365 – delete the letter a before issued.
Page 11 Line 466 – replace the word to with for or in, or to treat.
Page 11 Line 479 – insert the letter a before the word human.
Page 11 Line 495 – delete the letter a before high protein.
Page 12 Line 514 – insert a space between Pubmed and.
Page 12 Line 515 – delete the.
Page 12 Line 531 – if there are two periods after ongoing, please delete one of them. (Table 1 Title).
For Table 1, please move Clinical Trial Identifiers (NCT#’s) to include under the Disease column for Batten Disease (Page 12), Page 13 MLD, MPSI, MPSII, MPSIIIA, MPSIIIB, Multiple Systems Atrophy, Krabbe Disease, MPSII, Canavan Disease, Parkinson’s Disease, GM1, for consistency in the Table (like on Page 14).
Comments on the Quality of English Language
Great paper. Language looks fine, just a couple of really minor edits recommended.
